# The impact of COVID-19 on opioid toxicity deaths for people who experience incarceration compared to the general population in Ontario: A whole population data linkage study

**Amanda Butler**[1,2]\*, **Ruth Croxford**[3], **Katherine E. McLeod**[2], **Tara Gomes**[4,5,6], **Aaron M. Orkin**[7], **Susan J. Bondy**[8], **Fiona G. Kouyoumdjian**[2]

**1** School of Criminology, Simon Fraser University, Burnaby, British Columbia, Canada, **2** Department of Family Medicine, McMaster University, Hamilton, Canada, **3** Freelance Statistician, Toronto, Ontario, Canada, **4** MAP Centre for Urban Health Solutions, Li Ka Ching Knowledge Institute, St. Michael's Hospital, Toronto, Ontario, Canada, **5** Leslie Dan Faculty of Pharmacy, University of Toronto, Toronto, Ontario, Canada, **6** Unity Health Toronto, Toronto, Ontario, Canada, **7** Department of Family and Community Medicine, University of Toronto, Toronto, Ontario, Canada, **8** Dalla Lana School of Public Health, University of Toronto, Toronto, Ontario, Canada

\* albutler@sfu.ca

## Abstract

### Background

To inform preparedness and population health action, we need to understand the effects of COVID-19 on health inequities. In this study, we assess the impact of COVID-19 on opioid toxicity deaths among people who experience incarceration compared to others in the general population in Ontario, Canada.

### Methods

We conducted a retrospective cohort study for the period of January 1, 2015 to December 31, 2020. We accessed and linked coronial data on all opioid toxicity deaths in Ontario with correctional data for people aged 18 years and older who were incarcerated in a provincial correctional facility. We used data from the Statistics Canada Census to calculate whole population rates. We used an interrupted time series design and segmented regression to assess for change in the level or rate of increase in deaths due to opioid toxicity coinciding with the COVID-19 pandemic. We compared the impact of COVID-19 on the opioid toxicity death rates for people exposed and not exposed to incarceration.

### Results

Rates of opioid toxicity death increased with a linear positive slope in both persons exposed to incarceration and those not exposed over the study period. The start of COVID-19 measures coincided with a marked upward shift in the trend lines with modification of the effect of COVID-19 by both sex and exposure to incarceration. For persons exposed to

**Data Availability Statement:** Legal data sharing agreements prohibit the investigators from making the dataset publicly available (even if personal identifiers are removed). The data that the authors linked are in the custody of the Ontario Ministry of the Solicitor General and the Office of the Chief Coroner of Ontario, and are subject to statutory confidentiality requirements. In order to access the linked dataset, any interested persons would need to obtain permission from the Ministry of the Solicitor General (solgenresearch@ontario.ca) and from the Office of the Chief Coroner (Andrew.Stephen@ontario.ca), and should contact the project Principal Investigator, Dr Fiona Kouyoumdjian (kouyouf@mcmaster.ca) who could facilitate communication with the appropriate parties and the Hamilton Integrated Research Ethics Board.

**Funding:** Funding support was provided by the Canadian Institutes of Health Research through the Canadian Research Initiative in Substance Misuse (SMN-139150). The funders had no role in study design, data collection and analysis, decision to publish, or preparation of the manuscript.

**Competing interests:** The authors have declared that no competing interests exist.

incarceration, the risk ratio (RR) was 1.50 (95%CI 1.35–1.69) for males and 1.21 (95%CI 1.06–1.42) for females, and for persons not exposed to incarceration, the RR was 1.25 (95%CI 1.13–1.38) for males and not significant for females.

## Conclusions

COVID-19 substantially exacerbated the risk of opioid toxicity death, impacting males and females who experienced incarceration more than those who had not, with an immediate stepwise increase in risk but no change in the rate of increase of risk over time. Public health work, including pandemic preparedness, should consider the specific needs and circumstances of people who experience incarceration.

## Introduction

Mortality risk is substantially greater for adults who experience incarceration than for others in the general population [1, 2]. The days and weeks immediately after release from prison are particularly hazardous, with very high rates of death from drug toxicity [1, 3–8]. People transitioning from prison to community face significant challenges including a lack of continuity of health care, criminal record-related impediments to employment, a lack of affordable housing and transportation, and difficulty navigating a complex system of government and community-based services [9, 10]. Coupled with decreased drug tolerance that commonly occurs during incarceration [11, 12] and in the context of a toxic illicit drug supply, these challenges likely contribute to the increased risk of fatal drug poisoning among people leaving prison.

The drug toxicity crisis has worsened over the past two decades and has disproportionately impacted people who experience incarceration [7, 8, 13]. Since 2016, the crisis has been largely fuelled by illicitly manufactured fentanyl and fentanyl analogues and an unregulated drug supply which is unpredictable in composition [14]. The emergence of COVID-19, which was declared a pandemic by the World Health Organization in March 2020, has been associated with further exacerbation of substance use and drug-related deaths in North America and Europe [14–18] through several mechanisms [19]. While there is a lack of primary data supporting causal linkages between COVID-19 and increases in deaths by drug toxicity, theoretical mechanisms have been proposed based on available evidence [20]. For example, pandemic-related restrictions including physical distancing measures and reduced capacity and hours of service posed challenges to direct service provision for people who use drugs [21, 22]. Other pandemic-related factors that may have contributed to drug-related deaths include stress, social isolation, worsening mental illness and feelings of despair, reductions in staff resources due to reallocation to the pandemic response, and border restrictions that contributed to a more volatile unregulated drug supply [15, 20]. As a consequence of the COVID-19 pandemic, there is a greater likelihood of adulteration, falsification, and substitution of synthetic opioids, which may increase the risk of unintentional overdose [23]. Within correctional facilities, measures to mitigate transmission of COVID-19 included increased time in cells, restrictions on programming and care access, and limited visits and social interactions [24], all of which may worsen mental health and negatively impact transitions to the community.

While increased drug toxicity deaths coinciding with the pandemic have been observed across North America, we lack data on the impact of the pandemic on people who experience incarceration. We identified only one relevant observational study, which compared overdose

rates three months after release from jail among people with opioid use disorder in Massachusetts before and during the pandemic [25]. That study found that people released in the first six months of the pandemic experienced three times higher odds of overdose mortality than people released in the six months preceding (odds ratio = 3.06, 95%CI 1.49–6.26) [25]. The current study aims to expand on these findings by estimating the impact of the COVID-19 pandemic and related restrictions on opioid toxicity mortality in custody and post-release for people who experienced incarceration using a whole population cohort over a 6-year period, and to compare with people who did not experience incarceration.

## Materials and methods

The Ontario Ministry of the Solicitor General operates all provincial correctional institutions in Ontario (*i.e.*, facilities for adults who are on remand awaiting trial or sentencing, or who have been sentenced to less than two years in custody) and oversees and delivers health care in these facilities. Persons sentenced to two years or more in custody are transferred from provincial facilities to federal penitentiaries.

We accessed data from the Ministry of the Solicitor General ("correctional data") for all persons aged 18 and older who were detained or incarcerated in an Ontario provincial correctional facility between January 1, 2015 and December 31, 2020, including name, date of birth, sex, Indigenous identity, marital status, race, periods of admission to and release from custody, and reasons for release. In this paper, we use "incarcerated" to describe persons who were detained or incarcerated in provincial correctional facilities.

We accessed data from the Office of the Chief Coroner of Ontario ("coronial data") for all deaths from opioid toxicity between January 1, 2015 and December 31, 2020, including name, date of birth, sex, and date of death. In Ontario, there is a legal obligation to report any death from opioid toxicity to a coroner. Opioid toxicity deaths include all sudden and unexpected deaths where an opioid was identified in post-mortem toxicology and was determined to have directly contributed to the cause of death, either alone or in combination with other substances.

We linked coronial data for people who died from opioid toxicity between 2015 and 2020 to correctional data for people who experienced incarceration over this same period, using name, date of birth and sex, using Link Plus version 2.0 [26]. Link Plus produces all possible matches and assigns a score, with higher scores indicating a more likely match. The scores are based on how well each of the elements (first, middle and last name, date of birth and sex) match. The study team reviewed specific matches and came to consensus regarding whether to consider them valid. After we completed data linkage, we used deidentified data for all subsequent analyses.

Data for the whole population were based on Ontario data from the Canadian census, conducted in 2016 and 2021 [27], with annual population estimates determined using linear interpolation by age and sex.

Our *exposed* group was adults who were incarcerated for any period in an Ontario provincial correctional facility between 2015 and 2020, and our *unexposed* group was all other adults in the general population in Ontario. Persons entered the exposed group from the date of their first incarceration during the study period and contributed person time to the exposed group up to the end of the study or death, whether they remained in custody or were released to the community. We use the term "people exposed to incarceration" to describe the exposed group. Our *unexposed* group was everyone in the general population after removing people who had been exposed to provincial incarceration during the study period.

## Analyses

For those exposed to incarceration, we calculated descriptive statistics for age, marital status, cumulative time in custody, length of most recent incarceration and number of total corrections admissions during the period under study, stratified by sex.

To determine whether COVID-19 impacted people exposed to incarceration differently from the unexposed group, we used an interrupted time series (ITS) design, a quasi-experimental approach for evaluating effects of interventions that occurs at a specific point in time [28]. In an ITS study, data collected at multiple time points before and after an intervention or event are used to estimate the effects of the intervention or event [29]. In segmented regression, the change in intercept and/or slope pre- to post-intervention is used to test a causal hypothesis about the intervention [28]. For the current study, segmented regression was used to determine whether there was a change in level and/or slope in the rate of opioid-related deaths pre-post COVID-19, and to compare rates of death and the COVID-19-related changes in rates of death among people exposed to incarceration to the rates and changes in rates among people not exposed to incarceration.

The number of opioid-related deaths and time at risk were calculated by month. Rate of death was calculated as the number of deaths per 10,000 person years at risk. For the ITS analyses, COVID-19 was treated as an event which occurred on April 1, 2020. Pandemic measures such as the closure of nonessential businesses began on March 23, 2020 [30], and so April is the first complete month in which restrictions were in place. Opioid-related deaths were allocated to age groups based on the deceased person's age at time of death using the date of birth in the coronial data.

We conducted a multivariable negative binomial regression with a robust estimate of the error variance to model the number of opioid toxicity deaths over the study period. The explanatory variables were age group (18–24, 25–39, 40–49 and 50+), sex (male, female), population (exposed to incarceration, not exposed), and month (from January 2015 to December 2020). Two additional variables were included in the regression: COVID-19 (a binary variable which is 0 until March 2020 and 1 starting in April 2020, allowing the *level* of the rate to change), and a variable, equal to 0 up to March 2020 and equal to the number of months since COVID-19 restrictions, starting with April 2020 = 1, which allowed the *slope* to change over time following the imposition of pandemic-related restrictions. Interactions allowed for the possibility that trends in death rate would vary by age, sex, and population, and that the impact of COVID-19 would differ by age, sex, and population. There is one observation for each combination of age, sex, population and month. The outcome variable was the number of deaths for each observation. The natural log (ln) of time at risk for incarceration was the offset variable. The rates were checked for autocorrelation, checking separately by sex and exposure to incarceration, up to a lag of 12, which would allow for a seasonality effect.

We also performed a segmented Cox proportional hazards survival analysis using the counting style approach to look at whether death rates within the exposed group (those who experience incarceration) changed based on period post-release, whether that association changed over time, and whether the association changed with COVID-19. Each person's history was divided into short time segments. A new time segment was started each time the person transitioned in or out of custody. While the person was either incarcerated or in the community, a new time segment was started every 30 days. Each time segment for a given person was characterized by their sex, age at the start of the time segment, incarceration/community status at the start of the time segment, and by the month in which the time segment starts, from January 2015 to December 2020. Incarceration/community status was first categorized as either 'in custody' or 'in community,' with the 'in community' group being further categorized

into four groups by the number of days since their last incarceration: days 1–30; days 31–200; days 201–365; and more than 1 year in community. An interaction between the effect of COVID-19 and incarceration/community status (considering all five groups defined above) allowed the impact of COVID-19 to differ depending on whether the person was in custody or in one of the periods after release from custody.

Analyses were performed in SAS version 10.0 [31]. Two-tailed p-values less than 0.05 were considered to indicate statistical significance.

We obtained ethics approval from the Hamilton Integrated Research Ethics Board (#5878), as well as study approval from the Ministry of the Solicitor General and the Office of the Chief Coroner.

## Results

Available sociodemographic and correctional data are shown in Table 1 for people exposed to incarceration, and show that 85.7% were male, the median age at first incarceration during this period was 33 years (IQR = 25–43), the median cumulative days in custody from 2015 to 2020 was 18.8 for females and 48.6 for males, and the median length of the most recent incarceration was 9.0 days for females and 19.2 days for males. Overall, 58.7% of opioid toxicity deaths among the exposed group occurred within a year of release from custody, while 2.5% occurred during incarceration, and 38.8% occurred more than a year after release.

**Table 1. Characteristics of people exposed to incarceration in Ontario provincial correctional facilities between January 2015 and December 2020.**

| Characteristics | Females N = 18,146 | Males N = 110,629 | Total N = 129,152 |
|---|---|---|---|
| **Age at first admission or January 1, 2015** | | | |
| Median (IQR) | 32 (25–41) | 33 (26–44) | 33 (25–43) |
| 18–24 | 4,185 (23.06%) | 23,876 (21.58%) | 28,180 (21.82%) |
| 25–39 | 8,984 (49.51%) | 49,416 (44.67%) | 58,576 (45.35%) |
| 40–49 | 3,169 (17.46%) | 20,379 (18.42%) | 23,597 (18.27%) |
| 50+ | 1,808 (9.96%) | 16,955 (15.33%) | 18,796 (14.55%) |
| Missing | 0 | 3 (0.00%) | 3 (0.00%) |
| **Marital status** | | | |
| Married, common law | 3734 (20.58%) | 26,276 (23.93%) | 30,256 (23.43%) |
| Single, separated, divorced or widowed | 13,794 (76.02%) | 78,574 (71.02%) | 92,684 (71.76%) |
| Missing | 618 (3.41%) | 5,579 (5.04%) | 6,212 (4.81%) |
| **Cumulative days in provincial custody 2015–2020** | | | |
| Mean (SD) | 78.98 (149.12) | 159.84 (269.57) | 148.71 (258.11) |
| Median (IQR) | 18.80 (3.31–90.19) | 48.62 (6.01–183.57) | 41.90 (5.29–166.56) |
| **Length of most recent incarceration (days)** | | | |
| Mean (SD) | 40.37 (92.88) | 80.28 (164.91) | 74.69 (157.45) |
| Median (IQR) | 8.96 (2.20–38.57) | 19.18 (3.55–86.03) | 18.01 (3.15–78.59) |
| **Number of incarceration episodes 2015–2020** | | | |
| Mean (SD) | 2.4 (2.7) | 2.5 (2.7) | 2.5 (2.7) |
| Median (IQR) | 1 (1–3) | 1 (1–3) | 1 (1–3) |
| **Deaths/Death rate** | | | |
| Deaths from opioid toxicity (n) | 345 | 1853 | 2207 |
| Death rate (per 100 PYs at risk) | 1.90 | 1.68 | 1.71 |

Abbreviation: IQR, interquartile range; SD, standard deviation

There was no evidence of autocorrelation in the multivariable negative binomial regression model. In this model (Table 2), age was independently associated with risk of death from opioid toxicity, with those aged 25–39 and 40–49 years at increased risk compared to those aged 18–24. The interaction between sex and exposure was such that both males and females who were exposed to incarceration were at greater risk of opioid toxicity death than males and females not exposed to incarceration before COVID-19 (rate ratio (RR) for exposed males was 24.73 (95%CI 22.84–26.77)), and RR for exposed females was 66.79, (95%CI 59.26–75.29). Further, among those exposed to incarceration, females were at higher risk of opioid toxicity death than males (RR 1.32, 95%CI 1.09–1.59); however, among those not exposed to incarceration, the inverse was true (RR 0.49, 95%CI 0.42–0.58). There was a significant increase in risk per month following January 2015, but no interaction between the period of COVID-19 and month (p = 0.60), indicating no change in the slope associated with COVID-19 restrictions. However, the level changed significantly, and furthermore the change in level depended on sex (p-value for the interaction < .0001) and exposure to incarceration (p-value for the interaction < .0001). COVID-19 affected the exposed more than the unexposed and males more than females, so that the pre-COVID-19 trends were accentuated. Specifically, the regression found no significant impact of COVID-19 for females not exposed to incarceration but an RR of 1.25 for males not exposed to incarceration (95%CI = 1.13–1.37). Among individuals exposed to incarceration, the RR was 1.21 (95%CI 1.05–1.40) for females and 1.50 (95%CI = 1.34–1.69) for males.

The panel in Fig 1 shows the risk of opioid toxicity death by age group, sex, and exposure to incarceration, compared with males aged 18–24 in the general population in January 2015. The impact of COVID-19 was largest for males exposed to incarceration and elevated their risk above the risk of similarly aged females exposed to incarceration across all age groups.

Among people exposed to incarceration, results of the survival analyses and segmented regression for deaths in the post-release period are presented in Table 3. People had the lowest risk while in custody and using the period in custody as the reference category, the risk in the community post-release was the highest during days 1–30, with a hazard ratio (HR) of 12.60 (95%CI = 9.40–16.88) and fell with increasing time spent in the community.

Over the study period, the risk for each of the four periods post-release from custody increased, and it increased by the same amount regardless of time spent in the community (p-value for an interaction between year and time in the community = 0.08). The HR for the annualized increase in deaths from 2015 onwards was 1.15 (95%CI = 1.09–1.20) for males and 1.34 (95%CI = 1.21–1.49) for females. The interaction between COVID-19 and incarceration/ community status was not significant (p = 0.46), which indicates that there is no evidence that the impact of COVID-19 differed by time spent in the community, or in custody. Similar to the negative binomial regression, our model did not suggest that the trend over time (i.e., slope) changed after COVID-19 began (p = 0.64).

The predicted HRs comparing the four periods post-release and calendar months to people in custody in January 2015 are shown graphically in Fig 2, with the reference group of males aged 18–24 in custody in January 2015.

## Discussion

We found that COVID-19 immediately and substantially exacerbated the risk of opioid toxicity death for people who experienced incarceration between 2015 and 2020, but did not change the rate of increased risk over time. We found significant differences in the increase in risk during the COVID-19 period between people exposed and not exposed to incarceration, as well by sex. Consistent with pre-pandemic studies [1, 3, 4, 8], females exposed to

**Table 2. Risk ratios from multivariable negative binomial regression for the effect of COVID-19 on the risk of opioid toxicity death, by age, sex and exposure to incarceration for people exposed to incarceration in Ontario provincial correctional facilities between January 2015 and December 2020.**

| Effect | Risk ratio | 95% confidence interval | p-value[a] |
|---|---|---|---|
| **Sex x age interaction** | | | 0.023[b] |
| **Effect of age, January 2015** | | | |
| Males 18–24 (reference age) | 1.0 | | |
| 25–39 | 1.77 | 1.46–2.15 | <0.0001 |
| 40–49 | 1.80 | 1.48–2.19 | <0.0001 |
| 50+ | 1.20 | 0.99–1.46 | 0.062 |
| Females 18–24 (reference age) | 1.0 | | |
| 25–39 | 1.55 | 1.24–1.94 | <0.0001 |
| 40–49 | 1.67 | 1.32–2.11 | <0.0001 |
| 50+ | 1.30 | 1.03–1.64 | 0.026 |
| **Sex x population interaction** | | | <0.0001[c] |
| Females (vs. males, reference), not exposed | 0.49 | 0.42–0.58 | <0.0001 |
| Females (vs. males, reference), exposed | 1.32 | 1.09–1.59 | 0.0037 |
| Males exposed vs males not exposed (reference) | 24.73 | 22.84–26.77 | <0.0001 |
| Females exposed vs. females not exposed (reference) | 66.79 | 59.26–75.29 | <0.0001 |
| **Trend (rate of change over time) x age (effect of 12 months)** | | | 0.015[d] |
| Age 18–24 | 1.15 | 1.10–1.20 | <0.0001 |
| Age 25–39 | 1.17 | 1.14–1.20 | <0.0001 |
| Age 40–49 | 1.13 | 1.10–1.17 | <0.0001 |
| Age 50+ | 1.10 | 1.07–1.14 | <0.0001 |
| **COVID-19 x sex interaction** | | | 0.0006[e] |
| **COVID-19 x population interaction** | | | 0.0033[f] |
| Effect of COVID-19 on males exposed | 1.50 | 1.34–1.69 | <0.0001 |
| on females exposed | 1.21 | 1.05–1.40 | 0.0100 |
| on males not exposed | 1.25 | 1.13–1.37 | <0.0001 |
| on females not exposed | 1.00 | 0.90–1.12 | 0.96 |

Not exposed = not exposed to incarceration between Jan 2015 –Dec 2020; exposed = exposed to incarceration between Jan 2015- Dec 2020. There was no evidence of a change in the time-trend after COVID-19 (by sex p = 0.77, by age p = 0.93, by population p = 0.15, overall p = 0.60). While the immediate impact of COVID-19 differed by sex and population, there was no evidence that it differed by age (p = 0.11).

[a]For each interaction, the first p-value is the p-value for the interaction effect. The remaining p-values are p-values for the specific comparison.

[b] The interaction between age and sex indicates that the rate ratios differed by sex for each age group (and that the rate ratios for each sex differed by age) initially (in January 2015).

[c] The interaction between sex and population indicates that the rate ratios comparing females to males depended on the population; and that the comparison of exposed to not exposed differed by sex.

[d] The interaction between trend (rate of change over time) and age indicates that the change in the rate of death over time differed by age group (and that the difference among the age groups changed over time).

[e] The interaction between COVID-19 and sex indicates that the impact of COVID-19 was different for males and females.

[f] The interaction between COVID-19 and population indicates that the impact of COVID-19 was different for exposed than for the not exposed group.

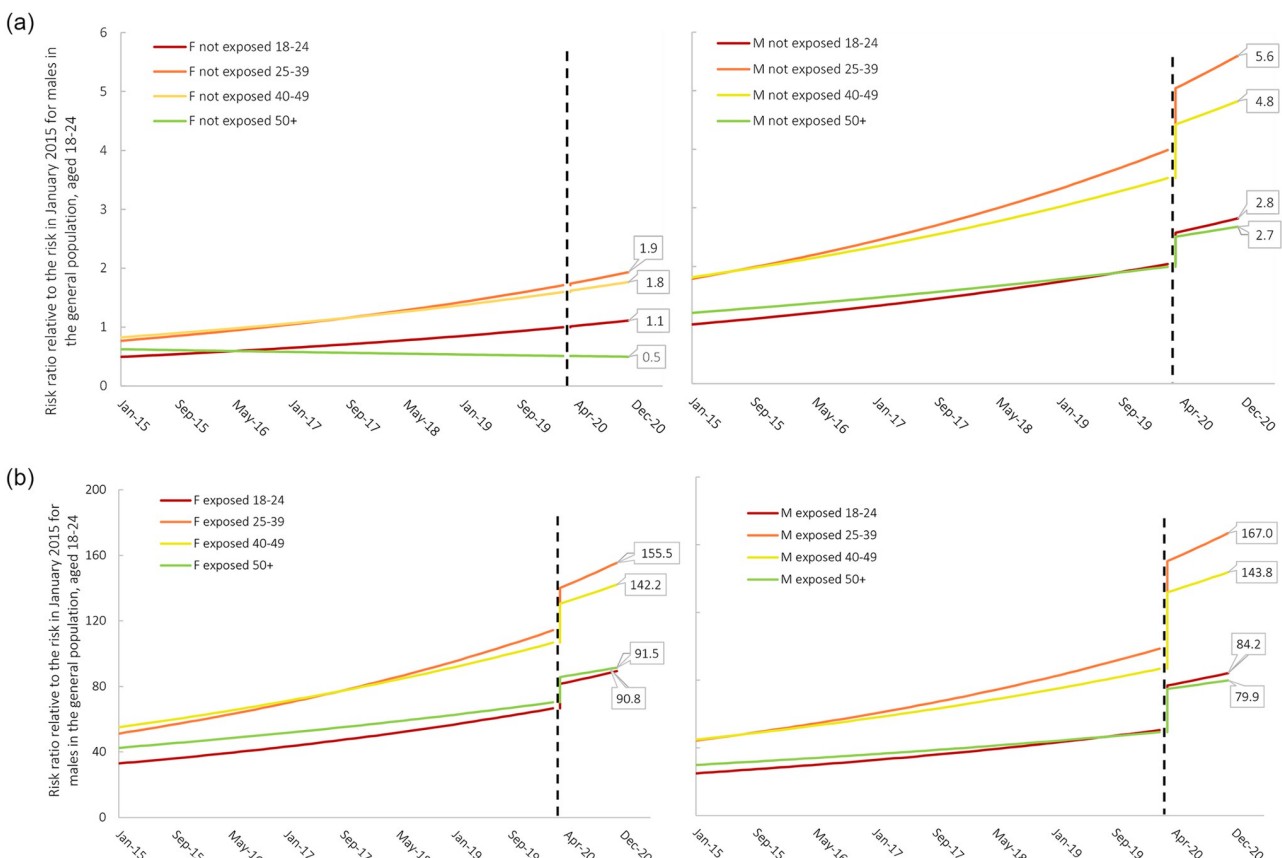

**Fig 1. Risk ratio for opioid toxicity death by sex, age group, and exposure to incarceration in provincial correctional facilities in Ontario between January 2015 and December 2020.** The dotted line represents the start of the first full month after COVID-19 restrictions were implemented (April 1, 2020). The Y axis units are the same for the not exposed groups (0–6) and for the exposed groups (0–200). F = female; M = male; not exposed = not exposed to incarceration between Jan 2015 –Dec 2020; exposed = exposed to incarceration between Jan 2015- Dec 2020.

incarceration continued to be at greater risk of death by opioid toxicity compared to females who were not exposed to incarceration. However, males exposed to incarceration were most significantly impacted by COVID-19 in terms of the increased risk of death. The highest risk period for death by opioid toxicity was in the first 30 days after release from custody. We did not identify a differential impact of the COVID-19 pandemic based on whether people exposed to incarceration were in custody or in the community post-release.

After the pandemic was declared, governments across Canada implemented a range of public health measures to limit the spread of the virus. By the end of March 2020, all Canadian provinces mandated closing universities, schools, public playgrounds and non-essential businesses [32]. Other measures included physical distancing, work-from-home requirements, limitations on the size of gatherings, canceling non-essential surgeries to increase hospital capacity, travel restrictions, testing, and contact tracing. The pandemic and response measures impacted trafficking of fentanyl and other drugs [21], while increases in drug prices, decreases in drug availability, and more drug adulteration were all documented since the start of the pandemic in Canada and in other jurisdictions [21, 33]. The pandemic, through mechanisms other than COVID-19 infection, substantially increased the risk of death by opioid toxicity across North America [15, 16, 19, 23]. This study contributes to emerging evidence that the

**Table 3. Hazard ratios from survival analyses for time to opioid toxicity death by age, sex and incarceration/community status over time and the impact of COVID-19 on time to death for people exposed to incarceration in Ontario provincial correctional facilities between January 2015 and December 2020.**

| Effect | Hazard ratio | 95% confidence interval | p-value[b] |
|---|---|---|---|
| **Age at the start of the month**[a] | | | <0.0001 |
| 18–24 (reference) | 1.0 | | |
| 25–39 | 1.79 | 1.52–2.12 | <0.0001 |
| 40–49 | 2.06 | 1.73–2.47 | <0.0001 |
| 50+ | 1.65 | 1.37–1.99 | <0.0001 |
| **Incarceration status at the start of the time period** | | | <0.0001 |
| Incarcerated (reference) | 1.0 | | |
| First 30 days in the community after release | 12.60 | 9.40–16.88 | <0.0001 |
| Days 31–200 in the community after release | 5.45 | 4.14–7.18 | <0.0001 |
| Days 201–365 in the community after release | 3.58 | 2.64–4.85 | <0.0001 |
| > 1 year in the community after release | 1.74 | 1.31–2.33 | 0.0002 |
| **Sex** | | | |
| Female (reference is male) in January 2015 | 0.70 | 0.45–1.08 | 0.11 |
| **Trend (how long after January 2015 did the time period start)** | | | 0.0066 |
| Trend x sex interaction | | | |
| Each additional 1 year, males | 1.15 | 1.09–1.20 | <0.0001 |
| Each additional 1 year, females | 1.34 | 1.21–1.49 | <0.0001 |
| **COVID-19 x sex interaction** | | | 0.041 |
| Effect of COVID-19, males | 1.52 | 1.30–1.76 | <0.0001 |
| Effect of COVID-19, females | 1.06 | 0.78–1.45 | 0.70 |

[a] In the counting process regression, a new time period was started when the individual transitioned from one status to another. While the individual remained in a given status a new time period was started every 30 days. The term "month" is being used as a shorthand since most time periods were 30 days in length.

[b] For each interaction, the first p-value is the p-value for the interaction effect. The remaining p-values are p-values for the specific comparison.

Two of the significant interactions seen in the negative binomial regression were not significant here: sex x age, p = 0.82 and trend x age, p = 0.44. There was no evidence that the trend over time differed by status, p = 0.08; the risk of an opioid-toxicity related death increased equally over time for all statuses, though it increased more for females than for males. There was no evidence that the trend over time (the slope) changed after COVID-19, p = 0.64.

pandemic further increased the already substantially elevated risk of death by opioid toxicity specifically for people who experience incarceration. The complex interplay between COVID-19 and opioid use has important implications for efforts to prepare for future public health emergencies, as well as ongoing responses to the lasting population and public health effects of the COVID-19 pandemic.

Our study has several strengths and limitations. We used whole population data, and given legally mandated reporting of deaths to coroners, we expect that the study has a high level of outcome ascertainment. Our outcome of interest is opioid toxicity deaths but for many of these deaths, other substances may have contributed to death (i.e., synergistic effects of different substances). We defined our exposure based on incarceration in provincial correctional facilities between the specific period of 2015 to 2020, and the ITS analyses included people who had an incarceration at *any point* in this six-year period. The mechanisms associated with incarceration history and risk of death from opioid toxicity are complex, as the population of people who experience incarceration faces substantial health and social inequities that elevate their risk of death, including childhood adversity and trauma, poverty, racism, low levels of education, under/unemployment, homelessness and precarious housing [34–36]. Concurrent use of multiple drugs has also been identified as a key risk factor for drug-related deaths in the

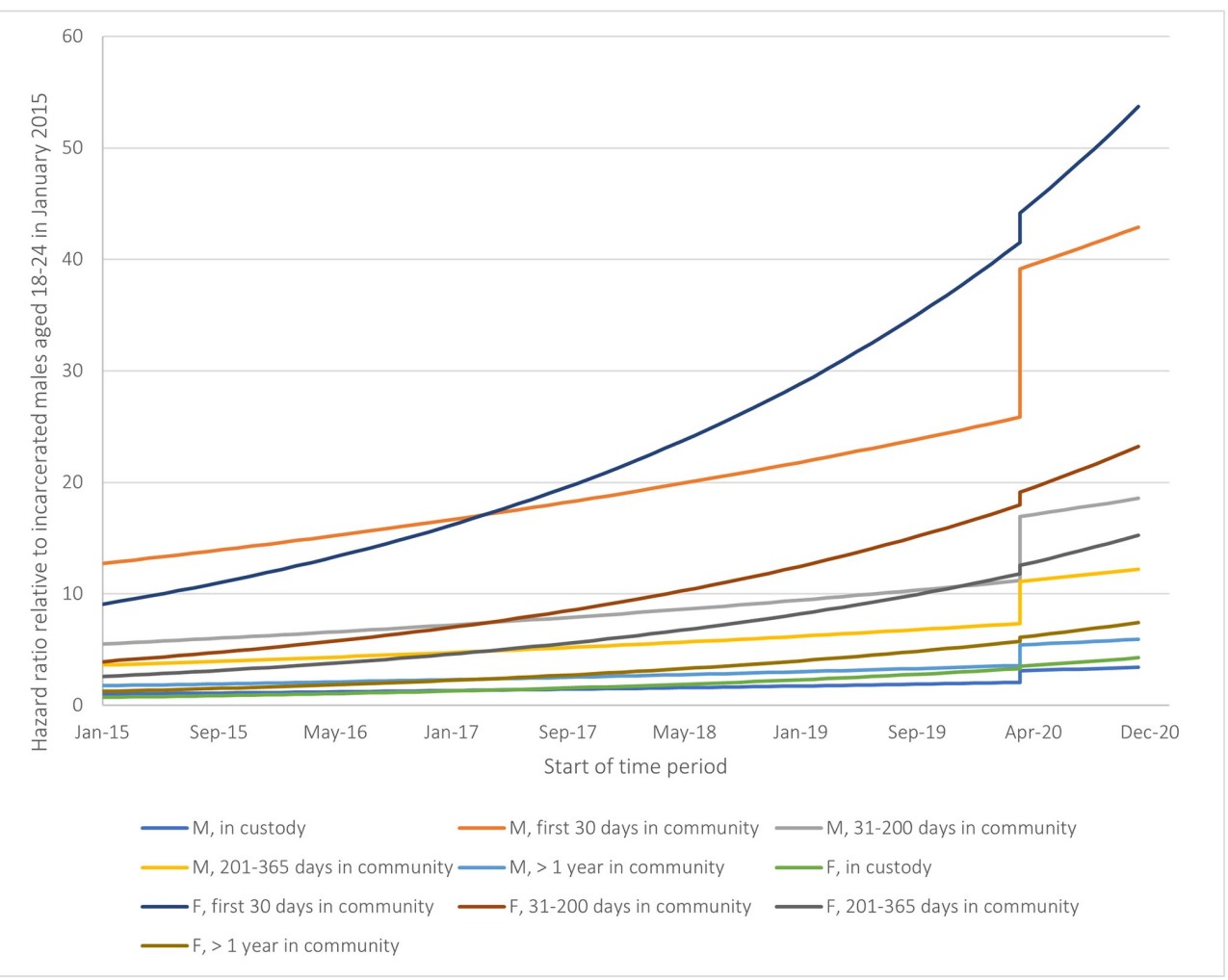

**Fig 2. Hazard ratios per month for deaths by opioid toxicity over time stratified by incarceration/community status for people who experienced incarceration in provincial correctional facilities in Ontario between January 2015 and December 2020.**

general population and limited research has found more common and more extensive poly-substance use patterns among people with criminal justice system involvement [37, 38].

We did not describe available data on race and Indigenous identity, as we did not have necessary partnerships in place to do so [39]. Our study includes only the first year of the pandemic and future research is required to determine the longer-term impacts of pandemic measures, which evolved and fluctuated over time. Finally, qualitative research is required to understand the experiences of people exposed to incarceration and the specific mechanisms of increased risk of adverse outcomes related to the pandemic and prison.

People who experience incarceration are at substantially greater risk of death by opioid toxicity compared with others in the general population. Increased deaths during the COVID-19 pandemic may be attributable to pandemic-related restrictions in services for people who use drugs [21, 40], as well as changes in conditions in custody such as greater isolation and less discharge planning, and in the community such as worse housing access and an increasingly toxic drug supply [19]. Additional research is needed to look all substances contributing to overdose deaths, which may be differential for those exposed to incarceration compared to

those who are not exposed. There is an urgent need to monitor and test the drug supply and drug exposures through drug checking, testing of seized drugs, and toxicology for people on relevant treatments [41], and to inform individual and population-level strategies (e.g., education and resources for people who use drugs, toxic drug alerts, increasing availability of supervised drug use settings, and strategies to address the toxic drug supply including pharmaceutical alternatives). As part of a broader public health agenda to address the health harms of incarceration, future emergency preparedness efforts should explicitly consider the prison setting and people who experience incarceration [19].

## Acknowledgments

We wish to acknowledge the Ontario Office of the Chief Coroner and Ministry of the Solicitor General for providing data for this study and for supporting data linkage.

## Author Contributions

**Conceptualization:** Amanda Butler, Katherine E. McLeod, Tara Gomes, Aaron M. Orkin, Susan J. Bondy, Fiona G. Kouyoumdjian.

**Data curation:** Ruth Croxford, Fiona G. Kouyoumdjian.

**Formal analysis:** Ruth Croxford.

**Investigation:** Amanda Butler.

**Methodology:** Amanda Butler, Ruth Croxford, Katherine E. McLeod, Tara Gomes, Susan J. Bondy, Fiona G. Kouyoumdjian.

**Resources:** Fiona G. Kouyoumdjian.

**Supervision:** Fiona G. Kouyoumdjian.

**Visualization:** Amanda Butler.

**Writing – original draft:** Amanda Butler.

**Writing – review & editing:** Amanda Butler, Katherine E. McLeod, Tara Gomes, Aaron M. Orkin, Susan J. Bondy, Fiona G. Kouyoumdjian.

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
