## [Decision Letter · Decision Letter 0]

17 Aug 2023

PONE-D-23-18165The impact of COVID-19 on opioid toxicity deaths for people who experience incarceration compared to the general population in Ontario: A whole population data linkage studyPLOS ONE

Dear Dr. Butler,

Thank you for submitting your manuscript to PLOS ONE. After careful consideration, we feel that it has merit but does not fully meet PLOS ONE’s publication criteria as it currently stands. Therefore, we invite you to submit a revised version of the manuscript that addresses the points raised during the review process.

We look forward to receiving your revised manuscript.

Kind regards,

Simona Zaami

Academic Editor

PLOS ONE

Journal Requirements:

Reviewers' comments:

Reviewer's Responses to Questions

**Comments to the Author**

1. Is the manuscript technically sound, and do the data support the conclusions?

Reviewer #1: Yes

Reviewer #2: Yes

2. Has the statistical analysis been performed appropriately and rigorously? 

Reviewer #1: Yes

Reviewer #2: Yes

3. Have the authors made all data underlying the findings in their manuscript fully available?

Reviewer #1: Yes

Reviewer #2: Yes

4. Is the manuscript presented in an intelligible fashion and written in standard English?

Reviewer #1: Yes

Reviewer #2: Yes

5. Review Comments to the Author

Reviewer #1: Dear Authors,

I have been tasked with reviewig the manuscript titled The impact of COVID-19 on opioid toxicity deaths for people who experience incarceration compared to the general population in Ontario: A whole population data linkage study.

I have certainly appreciated the depth and thoroughness the authors managed to achieve in terms of the stated objective. The article has an element of novelty that makes it a relevant and valuable contribution to toxicological research. In addition, the manuscript is well assembled and competently structured. The well explicated methodology is also a plus.

It would be advisable to lay out a higher degree of contextualization as to the COVID-19 pandemic and substance use, since such highly complex dynamics and trends are not comprehensively outlined in the article. Furthermore, how synthetic opioids and NPS use was impacted by the pandemic may also be worthy of analysis in the discussion. The following sources should be drawn upon and cited:

doi: 10.3389/fphar.2019.00563.

doi: 10.1016/j.bja.2020.07.004.

doi: 10.1002/hup.2727.

doi: 10.1016/j.euroneuro.2021.09.002.

doi: 10.3390/biology12020273.

The artcle is well-written and offers an insightful perspective and analysis.

Best regards.

Reviewer #2: I was called upon the review the submission by the title The impact of COVID-19 on opioid toxicity deaths for people who experience incarceration compared to the general population in Ontario: A whole population data linkage study.

The chief quality of the article I feel like pointing out is originality: the analysis comparing inmates to the general population is insightful; in addition, the methodological soundness the authors managed to achieve is noteworthy. The figures and tables have been well-crafted and are effective and straightforward in conveying key data and findings.

Certainly, I feel it would be advisable to provide a broader analytical perspective in the Discussion section, by briefly expounding upon opioids use and how screening and detection play a key role in tackling trafficking and abuse trends, also in reference to polydrug use, in prison vs general polulations.

The following sources can be looked at and added to the references:

doi: 10.3390/biology11050645.

doi: 10.1002/anie.202101262.

doi: 10.1093/jat/bks063.

The article is well written praiseworthy, especially as it relates to its stated goal.

I look forward to an improved revised version soon.

6. PLOS authors have the option to publish the peer review history of their article (what does this mean?). If published, this will include your full peer review and any attached files.

Reviewer #1: No

Reviewer #2: No

---

## [Author Response · Author response to Decision Letter 0]

3 Oct 2023

[See attached file with detailed responses to reviewer comments.]

We thank the reviewers for their kind comments and helpful suggestions. Our responses are provided below.

Reviewer #1: Dear Authors,

- I have been tasked with reviewing the manuscript titled The impact of COVID-19 on opioid toxicity deaths for people who experience incarceration compared to the general population in Ontario: A whole population data linkage study.

I have certainly appreciated the depth and thoroughness the authors managed to achieve in terms of the stated objective. The article has an element of novelty that makes it a relevant and valuable contribution to toxicological research. In addition, the manuscript is well assembled and competently structured. The well explicated methodology is also a plus.

Response: Thank you for your positive feedback. 

- It would be advisable to lay out a higher degree of contextualization as to the COVID-19 pandemic and substance use, since such highly complex dynamics and trends are not comprehensively outlined in the article. Furthermore, how synthetic opioids and NPS use was impacted by the pandemic may also be worthy of analysis in the discussion. The following sources should be drawn upon and cited:

doi: 10.3389/fphar.2019.00563.

doi: 10.1016/j.bja.2020.07.004.

doi: 10.1002/hup.2727.

doi: 10.1016/j.euroneuro.2021.09.002.

doi: 10.3390/biology12020273.

Response:

In the introduction, we discuss the complex dynamics associated with the COVID-19 pandemic and substance use. For reference, we provide the following content: 

“While there is a lack of primary data supporting causal linkages between COVID-19 and increases in deaths by drug toxicity, theoretical mechanisms have been proposed based on available evidence (19). For example, pandemic-related restrictions including physical distancing measures and reduced capacity and hours of service posed challenges to direct service provision for people who use drugs (20, 21). Other pandemic-related factors that may have contributed to drug-related deaths include stress, social isolation, worsening mental illness and feelings of despair, reductions in staff resources due to reallocation to the pandemic response, and border restrictions that contributed to a more volatile unregulated drug supply (15, 19). Within correctional facilities, measures to mitigate transmission of COVID-19 included increased time in cells, restrictions on programming and care access, and limited visits and social interactions (22), all of which may worsen mental health and negatively impact transitions to the community.”

We appreciate that recommendation to include papers about novel psychoactive substances (NPS). In the introduction we acknowledge that fentanyl and other synthetic opioids have dominated the illicit drug supply. We added Rinaldi et al. (2020) to the following sentence:

"The pandemic and response measures impacted trafficking of fentanyl and other drugs (19), while increases in drug prices, decreases in drug availability, and more drug adulteration were all documented since the start of the pandemic in Canada and other jurisdictions (20, 31)."

We added Pichini et al. (2020) to the following sentence: “The emergence of COVID-19, which was declared a pandemic by the World Health Organization in March 2020, has been associated with further exacerbation of substance use and drug-related deaths in North America and Europe (14-18) through several mechanisms (19)." 

We added the following sentence and cited Torrens and Fonseca (2022): “As a consequence of the COVID-19 pandemic, there is a greater likelihood of adulteration, falsification, and substitution of synthetic opioids, which may increase the risk of unintentional overdose (23).” 

For this paper, we are focusing specifically on opioid-related deaths (not the increased risk of death by COVID or other illnesses among people who use substances). As such, Lambert 2020 is not relevant.

- The article is well-written and offers an insightful perspective and analysis.

Response: Thank you. 

Best regards.

Reviewer #2: I was called upon the review the submission by the title The impact of COVID-19 on opioid toxicity deaths for people who experience incarceration compared to the general population in Ontario: A whole population data linkage study.

The chief quality of the article I feel like pointing out is originality: the analysis comparing inmates to the general population is insightful; in addition, the methodological soundness the authors managed to achieve is noteworthy. The figures and tables have been well-crafted and are effective and straightforward in conveying key data and findings.

Response: Thank you for your positive feedback. 

- Certainly, I feel it would be advisable to provide a broader analytical perspective in the Discussion section, by briefly expounding upon opioids use and how screening and detection play a key role in tackling trafficking and abuse trends, also in reference to polydrug use, in prison vs general populations.

The following sources can be looked at and added to the references:

doi: 10.3390/biology11050645.

doi: 10.1002/anie.202101262.

doi: 10.1093/jat/bks063.

Response:

We appreciate this feedback. Although we did not examine polysubstance use, we agree that this is an important risk factor for drug related deaths, particularly among incarcerated populations. We added the following to the discussion:

“Concurrent use of multiple drugs has also been identified as a key risk factor for drug-related deaths in the general population and limited research has found more extensive polysubstance use patterns among people with criminal justice system involvement (37, 38).”

The following has been added to the limitation section:

 “Our outcome of interest is opioid toxicity deaths but for many of these deaths, other substances may have contributed to death (i.e., synergistic effects of different substances).”

The following was added to the discussion section and we cited Kroning and Wang (2021):

“Additional research is needed to look all substances contributing to overdose deaths, which may be differential for those exposed to incarceration compared to those who are not exposed. There is an urgent need to monitor and test the drug supply and drug exposures through drug checking, testing of seized drugs, and toxicology for people on relevant treatments (41), and to inform individual and population-level strategies (e.g., education and resources for people who use drugs, toxic drug alerts, increasing availability of supervised drug use settings, and strategies to address the toxic drug supply including pharmaceutical alternatives).”

- The article is well written praiseworthy, especially as it relates to its stated goal.

I look forward to an improved revised version soon.

Response: Thank you.

---

## [Editor Report · Decision Letter 1]

10 Oct 2023

The impact of COVID-19 on opioid toxicity deaths for people who experience incarceration compared to the general population in Ontario: A whole population data linkage study

PONE-D-23-18165R1

Dear Dr. Butler,

We’re pleased to inform you that your manuscript has been judged scientifically suitable for publication and will be formally accepted for publication once it meets all outstanding technical requirements.

Kind regards,

Simona Zaami

Academic Editor

PLOS ONE

Additional Editor Comments (optional):

Dear Authors,

I believe you have successfully amended and improved the article overall, and I am confident the reviewers' requests and suggestions have been met.

in light of the article's strengths (relevance, novelty, thoroughness and coherence), I believe it should be approved for publication.

Best regards,

Prof. Simona Zaami

---

## [Editor Report · Acceptance letter]

16 Oct 2023

PONE-D-23-18165R1 

The impact of COVID-19 on opioid toxicity deaths for people who experience incarceration compared to the general population in Ontario: A whole population data linkage study 

Dear Dr. Butler:

I'm pleased to inform you that your manuscript has been deemed suitable for publication in PLOS ONE. Congratulations! Your manuscript is now with our production department. 

Kind regards, 

on behalf of

Dr. Simona Zaami 

Academic Editor

PLOS ONE